# Efficient Propagation of Circulating Tumor Cells: A First Step for Probing Tumor Metastasis

**DOI:** 10.3390/cancers12102784

**Published:** 2020-09-28

**Authors:** Jerry Xiao, Joseph R. McGill, Kelly Stanton, Joshua D. Kassner, Sujata Choudhury, Richard Schlegel, Zuben E. Sauna, Paula R. Pohlmann, Seema Agarwal

**Affiliations:** 1Lombardi Cancer Center, Georgetown University Medical Center, Washington, DC 20007, USA; jx109@georgetown.edu (J.X.); paula.r.pohlmann@gunet.georgetown.edu (P.R.P.); 2Department of Pathology, Center for Cell Reprogramming, Georgetown University Medical Center, Washington, DC 20007, USA; sc362@georgetown.edu (S.C.); Richard.Schlegel@georgetown.edu (R.S.); 3Hemostasis Branch, Division of Plasma Protein Therapeutics, Office of Tissues and Advanced Therapies, Center for Biologics Evaluation and Research, U.S. Food and Drug Administration, Silver Spring, MD 20993, USA; Joseph.McGill@fda.hhs.gov (J.R.M.); zuben.sauna@fda.hhs.gov (Z.E.S.); 4Department of Pathology, Yale University, New Haven, CT 06511, USA; kelly.stanton@yale.edu; 5Department of Medicine, Medstar Hospital, Georgetown University Medical Center, Washington, DC 20007, USA; jdk112@georgetown.edu

**Keywords:** Cancer metastasis, circulating tumor cells, tumor-associated neutrophils, leukocytes, cell culture

## Abstract

**Simple Summary:**

Cancer metastasis is responsible for most cancer-associated death. During metastasis, cells that escape the primary tumor into the circulatory system are known as circulating tumor cells. Previous attempts at growing circulating tumor cells in the lab have been hindered by low success rates. Using the novel system first reported here, we demonstrate a 100% (12/12 samples) success rate in culturing circulating tumor cells from metastatic breast cancer patients. Once propagated, we characterized the expression profiles of our cultures, verifying their origins as breast cancer cells. Furthermore, exploratory analysis identifies several key pathways and genes previously known to be associated with breast cancer progression and metastasis. Finally, we demonstrate that cultures grown in the presence of CD45^+^ cells exhibited higher growth potential ex vivo. Based on this system, we suggest that a reconsideration of the parameters for circulating tumor cell isolation should be undertaken.

**Abstract:**

Circulating tumor cells (CTCs) represent a unique population of cells that can be used to investigate the mechanistic underpinnings of metastasis. Unfortunately, current technologies designed for the isolation and capture of CTCs are inefficient. Existing literature for in vitro CTC cultures report low (6−20%) success rates. Here, we describe a new method for the isolation and culture of CTCs. Once optimized, we employed the method on 12 individual metastatic breast cancer patients and successfully established CTC cultures from all 12 samples. We demonstrate that cells propagated were of breast and epithelial origin. RNA-sequencing and pathway analysis demonstrated that CTC cultures were distinct from cells obtained from healthy donors. Finally, we observed that CTC cultures that were associated with CD45^+^ leukocytes demonstrated higher viability. The presence of CD45^+^ leukocytes significantly enhanced culture survival and suggests a re-evaluation of the methods for CTC isolation and propagation. Routine access to CTCs is a valuable resource for identifying genetic and molecular markers of metastasis, personalizing the treatment of metastatic cancer patients and developing new therapeutics to selectively target metastatic cells.

## 1. Introduction

Cancer metastasis is estimated to cause ~90% of cancer-associated deaths [1]; however, the mechanistic underpinnings of metastasis remain elusive. Circulating tumor cells (CTCs), a population of cells that enter the blood stream from the primary tumor and seed distant sites, play a central role in metastasis [2]. Single-cell mutational analyses have identified novel mutations in CTCs compared to primary tumor tissue [3,4]. These studies indicate that CTCs, if properly isolated and characterized, can be used to elucidate the genetic and molecular markers of disease progression and of metastasis. A reliable and robust method to culture CTCs from individual patients would also allow the evaluation of alternative treatment regimens and personalize therapy. The immense potential of using CTCs in advancing the understanding of metastasis and developing novel clinical interventions has been limited by the rarity of these cells as < 20 CTCs are obtainable from 10 mL of blood. Consequently, to obtain sufficient numbers of CTCs for meaningful basic science and translational studies, it is necessary to isolate this ultra-rare cellular subpopulation and to expand the CTCs in culture. Unfortunately, technologies to identify and propagate CTCs in culture remain inefficient [5,6,7,8]. 

Current methods to isolate CTCs rely on; (i) antibodies to CTC markers, or (ii) microfluidic manipulation based on physical properties of the cells. The pioneering CellSearch technology is the only FDA approved method for isolating CTCs [9]. CellSearch relies on an epithelial cell adhesion molecule (EpCAM)-positive and leukocyte common antigen (CD45)-negative selection process using antibodies to these markers. Thus, epithelial cells which are EpCAM^+^ are harvested while leukocytes which are CD45^+^ are removed [9]. The method has proved to be a useful analytical tool for enumeration of CTCs in patient blood and this measure has been effective in predicting patient treatment response [10], prognosis [11], and disease recurrence [12]. However, the method has had limited success in routinely culturing and expanding CTCs. While the likelihood of success varies based on the cancer and stage, CTCs from only approximately one-in-five patients can be successfully cultured. Recent studies, moreover, indicate that relying solely on a single epithelial marker fails to capture the heterogeneity of CTCs [8,13]. For instance, cells which play an important role in metastases but are not EpCAM^+^ would not be isolated by this method. 

Alternative methods to isolate CTCs are based on physical cellular properties such as cell density [14] and size [15,16]. The use of these methods expose fragile CTCs to mechanical-stresses and have consequently resulted in the same low success rates for culturing CTCs (6−20% of cultures initiated) [5,6,7] as the method using the epithelial marker EpCAM. 

While the lack of a method to reliably culture CTCs from most patients has remained a challenge, when such cultures are successfully established, they have provided valuable insights into the metastatic process and the therapeutic response of individual patients [5,6,7]. Therefore, a technology that allows reliable and consistent expansion of CTCs from individual patients and captures the heterogeneity of this cellular subpopulation remains a significant unmet need in cancer biology. 

We report a method to grow and expand CTCs. We used blood samples from 12 metastatic breast cancer (MBC) patients to successfully generate short-term cultures from all patients, compared to a lower success rate with currently available techniques [5,6,7]. Importantly, we have carried out extensive characterization of the cultured and expanded CTCs to demonstrate the epithelial and breast origin of the cells. RNA-sequencing (RNA-seq) analysis showed significant differences in the genetic signatures of CTCs cultured using our method and cells from healthy donors. We therefore fulfill a critical unmet need, namely CTCs in sufficient numbers to study the genetic determinants of metastasis and design patient-specific personalized treatment strategies.

## 2. Results

### 2.1. CTCs Are Propagated Using an Unbiased Selection Method

To expand CTCs, we leveraged a technology that is free of any selection bias and does not expose CTCs to undue mechanical stresses. For workflow and culture conditions, see Figure 1a and the Methods section. An initial isolation step separates red blood cells (RBCs) from CTC-containing plasma and the buffy coat using Ficoll-Paque density gradient. In contrast to previous studies, we harvested all remaining cells in the plasma and buffy coat along with the CTCs. The rationale for doing so is that several studies have shown that CD45^+^ leukocytes (contained in the buffy coat) favor CTC survival [17]. 

We applied the standardized method for culturing CTCs to peripheral blood samples from a cohort of 12 metastatic breast cancer (MBC) patients. The 12 MBC patients represent all three breast cancer subtypes (estrogen receptor (ER)/progesterone receptor (PR) positive *n* = 5; human epidermal growth factor receptor 2 (HER2) positive *n* = 3; triple-negative breast cancer (TNBC) *n* = 4; Figure 1b). As controls, we used peripheral blood samples from five healthy donors (HD). Importantly, RNA samples from HD were harvested from the buffy coat, as no HD cells were expected to survive under our culture conditions. Cells from MBC patients and HD were cultured under identical conditions; all 12 MBCs resulted in successful cultures while none of the HD samples resulted in viable cultures (Figure 1c, right panel). 

Interestingly, all twelve samples exhibited growth up to 30 days. Notably, we observed that 6 of the 12 CTC cultures exhibited higher growth potential, i.e., could be cultured for >30 days. These six cultures became adherent to the plates and were associated with CD45^+^ cells (Figure 1c, left panel, Figure 1d,e). This observation supports recent studies showing that CD45^+^ leukocytes support CTC survival and justifies our decision to include cells from the buffy coat in our culture. This observation also suggests that the deliberate exclusion of CD45^+^ leukocytes, in methods such as the CellSearch technology, may contribute to the low success rate for establishing successful CTC cultures.

### 2.2. Propagated CTCs Are of Breast Epithelial Origin

After 30 days, we harvested RNA from all 12 putative CTC cultures. To validate that the successfully propagated cultures contain CTCs, we used appropriate markers to establish the epithelial and breast origin of these cells. Cytokeratin 5 and 8 were chosen as luminal and basal epithelial markers of breast cancer, respectively. Additionally, mammaglobin, a member of the secretoglobin family and biomarker for breast cancer, was chosen to verify the breast origin of the cells. We demonstrated that the cultures established from all 12 MBC patients tested positive for cytokeratin 5 and 8 and mammaglobin using RT-PCR (Figure 1d). Notably, epithelial (cytokeratin 5) and breast (mammaglobin) markers did not amplify in RNA from healthy donors (Appendix A). RT-PCR for CD45 in healthy donors exhibited amplification, in concordance with the fact that some leukocytes may be present in the plasma and buffy coat layers following Ficoll-Paque processing (Appendix A).

We next sought to characterize gene expression of the expanded cells using RNA-seq. Only the 6 samples that could be propagated for over 30 days yielded sufficient quantities of RNA for analyses. These samples were the CTCs that associated with CD45^+^ leukocytes (Figure 1d,e); emphasizing the importance of an unbiased methodology that does not deplete leukocytes for culturing CTCs. RNA from blood samples from the five HDs were also sequenced to determine the background RNA-seq profile of leukocytes in the blood. 

A principle-component analysis of gene expression data shows that the 5 HDs are clustered together, and distinct from all 6 CTC samples (Figure 2a). CTCs demonstrated no obvious clustering based on receptor status of the patient from whom they were obtained. Overall, RNA-seq identified 7234 genes that were significantly (adjusted *p*-value < 0.01) differentially expressed in the CTCs compared to HDs. Of the 7234 genes that were differentially expressed, 3657 were upregulated and 3577 were downregulated (Figure 2b,c). Additionally, among the top upregulated genes, high expression of several genes resulted in significantly decreased overall survival when assessed in the METABRIC study (Appendix A).

To assess whether CTCs were enriched in the cultured cells, we examined a panel of genes associated with breast, epithelial, mesenchymal, and stem cells. Epithelial markers (KRT8; KRT18; CLDN7; CTNNB1) and mesenchymal markers (VIM; SERPINE1; SNAI1; ACTA2) were upregulated in CTC samples vis-à-vis HD samples (Figure 2d and Appendix A). Cancer stem cell markers ALDH1A2, ALDH7A1, CD44, and CCND1 were also upregulated in CTCs compared to HDs (Figure 2d). Additionally, a recent study compared the RNA-sequencing profiles of MBC-derived CTCs with HDs [18]. Our results are consistent with their findings providing further evidence that our technology propagates CTCs (compare Appendix A and Figure 1b in reference [18]).

### 2.3. CTCs Demonstrate Association with Cancer Pathways

To confirm that the putative CTCs, expanded using our culture conditions were of cancer origin, we analyzed our RNA-seq data using the Kyoto Encyclopedia of Genes and Genomes (KEGG) computational tool. We found that 52 pathways were significantly enriched (adjusted *p*-value < 0.05; Figure 3a and Appendix A) [19]. The most significantly enriched (adjusted *p*-value = 4.35 × 10^-21^) pathway was the KEGG pathway hsa01100 which encompasses all metabolic processes in the cell, consistent with the dysregulation of metabolic pathways in cancer cells [20]. KEGG analysis also detected other pathways more explicitly associated with cancer, including pathways in cancer (hsa05200) and transcriptional misregulation in cancer (hsa05202) (Figure 3a). Finally, KEGG analysis highlighted several signaling pathways associated with cancer progression and metastasis, including FoxO (hsa04068) [21], p53 (hsa04115) [22], and RAP1 (hsa04015) [23] (Figure 3a). 

Another powerful bioinformatics tool that is used to identify affected biological processes based on gene expression data is the Gene Ontology (GO) project [24,25]. Notably, biological processes such as positive regulation of locomotion (GO:0040017, FDR = 6.61 × 10^−9^) and cell migration (GO:0030335, FDR = 5.94 × 10^−9^) suggest that propagated CTCs exhibit migratory properties consistent with the properties of metastatic cells (Figure 3b and Appendix A). Furthermore, among the highly enriched processes were those associated with leukocyte activation (GO:0002274, FDR = 7.35 × 10^−18^), and neutrophil activation (GO:0042119, FDR = 1.10 × 10^−13^). The latter observation confirms (by an orthogonal method) our finding that leukocytes are present in our cultures.

As gene expression analyses that identify individual genes are not always successful at detecting multi-gene biological processes, the bioinformatics tool, Gene Set Enrichment Analysis (GSEA), focuses on gene sets [26]. Using GSEA, we identified 31 gene sets that were enriched in CTCs compared to HDs (Figure 3c and Appendix A). Gene sets associated with epithelial mesenchymal transition (normalized enrichment score (NES) = 2.03), early (NES = 1.60) and late (NES = 1.52) estrogen response, which are biological properties of breast cancer and metastasis, were enriched in CTCs compared to HDs (Figure 3d). In addition, mTORC1 (Figure 3e), TGF-β (Figure 3f), and p53 pathway (Figure 3g) gene sets were also enriched in CTCs and further validation of RNA-seq data using qRT-PCR and a subset of genes from these pathways resulted in a similar trend when compared to HD samples (Appendix A). Overall, the top 21 gene sets that were enriched (and were statistically significant) are important players in the tumorigenesis and metastasis process (Appendix A). 

### 2.4. CD45^+^ Cells Support CTC Growth In Vitro

As indicated above, recent evidence has suggested that CTCs associate with leukocytes in circulation [17,27]. Additionally, we have observed that CTCs associated with CD45^+^ cells result in longer term culture survival (and greater numbers of CTCs for experimental studies/clinical evaluation) (Figure 1c–e). To determine the identity of leukocytes in our cultures, we used two computational tools that allow the identification of immune cell types based on gene expression data: (i) Estimating the Proportion of Immune and Cancer cells (EPIC) [28] and (ii) CIBERSORT [29]. These two tools differ based on the source gene-expression data used to define the individual immune cell signatures. Using EPIC, a majority of cells in our CTC cultures classified as “other”; i.e., of non-immunological origin (Figure 4a). This finding is consistent with the expansion of CTCs (of epithelial origin) and depletion of blood cells (including immune cells) during our culture conditions (Figure 1b). However, when we analyzed only the immune cell populations, we found that NK cells (2.1% in CTCs vs. 10.3% in HDs) and CD4 T cells (38.5% in CTCs vs. 67.9% in HD) were depleted in the CTCs. Conversely the proportion of neutrophils (3.5% in CTCs vs. 0.00002% in HD), monocytes (20.9% in CTC vs. 3.6% in HD), and CD8 T cells (28.7% in CTCs vs. 10.4% in HD) increased in CTC cultures vs HD samples (Figure 4b). Notably, neutrophils showed the highest relative enrichment. Using CIBERSORT, a similar pattern emerges, i.e., CD8 T cells, neutrophils, and M2 macrophages are enriched in CTC samples (Figure 4c). Finally, ImSig [30], a third analytical tool used to identify immune cell sub-populations based on gene signatures also validated our findings by showing enrichment of genes associated with macrophages (Figure 4d) and neutrophils in CTC cultures (Figure 4e). 

## 3. Discussion

Current methods for the isolation of CTCs that rely on EpCAM-positive, CD45-negative selection [9] can enumerate CTCs but are not always successful at propagating CTCs. More recent studies demonstrated that both macrophages and neutrophils associate with CTCs [17,27,31,32]. Here we developed a technology for the short-term cultures of CTCs that exposes CTCs to minimal stress and is free of the biases introduced by a biomarker-based selection. We have demonstrated the reliability, reproducibility and robustness of our method by establishing successful cultures from blood obtained from 12 MBC patients. The putative CTCs cultures exhibited cellular markers associated with breast cancers (cytokeratin 5, cytokeratin 8, and mammaglobin), but are lacking in red blood cells which would be the only plausible contaminant [33]. RNA-seq analysis also showed that the CTC cultures cluster separately from HDs. Finally, the analyses of the RNA-seq data showed that genes and pathways upregulated in the putative CTCs are either associated directly with human cancers, the metastatic process, or cellular physiologies that are altered by cancers/metastasis. 

Our study highlighted another critical criterion for the longer-term viability of CTC cultures. The common characteristic of all CTC cultures that survived for >30 days was that they were associated with CD45^+^ leukocytes. We also unequivocally demonstrated, using three independent tools that identify immune cells in a sample based on RNA-seq data, that our technology expands and enriches CTCs and, among leukocytes, favors the survival of neutrophils and macrophages. Based on results from computational tools, a minority population (≤20%) of CTC cultures are likely derived from immune signatures (Figure 4). These leukocytes likely contribute to the success of our technology in establishing CTC cultures from individual breast cancer patients. Therefore, these results suggest, at a minimum, re-evaluating the strategy used in the FDA approved CellSearch technology for isolating CTCs [9]. This method includes depleting CD45^+^ leucocytes from patient blood samples during the process of isolating CTCs. The removal of CD45^+^ leucocytes likely contributes to the poor and inconsistent success in establishing cultures from CTCs isolated using CellSearch [5,6,7]. 

Although the purpose of our genetic analysis here was to demonstrate that the cultured cells were derived from CTCs, our results suggest interesting hypotheses which can be investigated in subsequent studies. For instance, among the highly upregulated genes in the putative CTC cells we identified BCAT1, a member of the mTORC1 signaling pathway. This gene has been implicated in tamoxifen resistance in breast cancers (Figure 3e). It would be intriguing to examine whether BCAT1 expression in CTCs correlates with anti-estrogen resistance in a clinical setting. If that is the case, CTCs, cultured using our method, could be used to obtain genetic information that guides therapeutic decisions. A similar pattern of expression can also be seen with TGF-beta signaling, which has previously been frequently linked to progression of breast cancer (Figure 3f) [34,35]. Furthermore, examination of the p53 signaling pathway reveals upregulated genes such as NUPR1, ATF3, and CDKN1A, all of which have known effects in breast cancer [36,37,38] (Figure 3g). 

Besides demonstrating the utility of cultured CTCs in identifying genes and pathways that have been previously associated with MBC we identified additional genes that are dysregulated in CTCs. There has been speculation in the literature on the role of many of these genes (e.g., PHLDA3 [39], SDC1 [40], and GM2A [41]) in tumorigenesis, but these genes were not identified in the context of MBC or CTCs. Access to sufficient numbers of CTCs would permit testing hypotheses pertaining to the role of these genes in tumor metastasis.

An intriguing observation in this study was a reversal in the ER/Her2 status in the CTCs compared to the primary tumor (Appendix A). Clinical studies have shown that such a switch in the ER/Her2 status occurs in 7−40% of patients when the breast cancer metastasizes [42,43,44]. Receptor switching poses an additional treatment challenge for patients. The ability to detect this switch through CTC analyses could abbreviate the need for serial biopsies to guide therapy. This is of particular relevance for patients in whom the metastatic sites are not accessible for sequential biopsies. 

There are some limitations to this study. Due to the limited amounts of blood that can be drawn from cancer patients and the rarity of CTCs, we cannot make head-to-head comparisons with other methods for isolating and expanding CTCs. There is however a critical mass of literature showing that CTCs isolated using existing methods result in low and inconsistent success rates with respect to establishing cultures. In addition, in this report we have focused only on CTCs from breast cancer patients. Thus, additional studies are required to demonstrate the broad suitability of the method across different cancers. Similarly, although the expansion of CTCs can provide sufficient numbers of CTCs for genetic and molecular to gain insights into the metastatic process. Currently, we are only able to establish short-term cultures. Finally, while we observed the presence of CD45^+^ cells in our cultures, there may be other cells present yet to be identified. Future studies should seek to clarify the identity of non-epithelial cells within the culture. Furthermore, future investigations may alternatively focus on the establishment of CTC derived xenografts (CDX) in animal models and the establishment of long-term cultures and individual patient-derived cell lines. 

## 4. Materials and Methods

### 4.1. Patient Enrollment

Patients were recruited, consented and enrolled at the MedStar Georgetown University Hospital Medical Oncology clinics in compliance with the Health Insurance Portability and Accountability Act (HIPAA) and Georgetown University Institutional Review Board (IRB) procedures (approval ID: MODCR00001156) through the Survey, Recruitment and Biospecimen collection Shared Resource (SRBSR) of the Lombardi Comprehensive Cancer Center. All patients provided written informed consent for the study. Inclusion criteria for patient enrollment included adult patients (>18 years old), male or female, with stage IV or recurrent/inoperable metastatic breast cancer (not oligometastatic) with no limit of prior treatment lines and with progression of disease since most recent therapy. All 12 samples were derived from female patients. Patients continued to be recruited until patients of the following histological categories had been consented: (1) HER2 positive, (2) TNBC, and (3) hormone receptor (HR: ER/PR) positive with a total of 12 patients. Exclusion criteria included any one or more of the following: (1) use of chemotherapy and/or antibody base therapy in the past 4 weeks; (2) use of radiotherapy in the past 2 weeks unless there are metastatic lesions beyond the radiated lesion. An additional 5 healthy donors with no known health conditions at the time of consent were also enrolled and used as controls for this study.

### 4.2. CTC Processing

Patients were enrolled and consented according to previously identified criteria. Once enrolled, an initial tube of blood was drawn and discarded, to prevent contamination of cultures from skin puncture. Two subsequent tubes of 7.5 mL blood each were drawn from each patient.

Blood samples were processed within 30 min of collection from patients. Samples were mixed with 1× Hank’s Balanced Salt Solution (HBSS) at 1:1 volume/volume ratio at room temperature. Samples were then split evenly into two 50 mL tubes containing 15 mL Ficoll-Paque (Cytiva Life Sciences, Marlborough, MA, USA cat no. 17-1440-02) each, being careful not to mix samples with Ficoll-Paque. Samples were spun for at 400× *g* for 40 min at 4 °C with minimal acceleration and deceleration, at which point plasma, buffy coat, Ficoll-Paque, and RBCs formed four distinct layers. The plasma and buffy coat layers were combined, harvested, and mixed with 1× HBSS to 50 mL final volume and spun at 400× *g* for 20 min at 25 °C. Steps taken after this were done under sterile conditions. Once spun, the supernatant was aspirated and two additional washes using 1× phosphate buffered saline (PBS) were performed. Washes were spun at 400× *g* for 10 min at 4 °C. Upon completion of the final wash, cells were resuspended in culture medium and plated for short-term cultures at 37 °C. The culture medium consisted of Advanced DMEM/F12 supplemented with B27 supplement (1×; Thermo Fisher, Grand Island, NY, USA, cat. no. 17504044), epidermal growth factor (10 ng/mL; Sigma-Aldrich, Saint Louis, MO, USA, cat. no. E9644), basic fibroblast growth factor (10 ng/mL; R&D systems, Minneapolis, MN, USA, cat. no. 233-FB), heparin (10 ug/mL; Sigma-Aldrich cat. no. H3149), Y-27632 (10 uM; Enzo Life Sciences, Farmingdale, NY, USA, cat. no. ALX-270-333), Adenine, L-Glutamine (2 mM; Thermo Fisher cat. no. 25030149), and Antibiotic-antimycotic (1×; Thermo Fisher cat. no. 15240062). Cultures were supplemented with fresh medium every 3 days and washed every 6 days with 1× PBS by centrifugation of the supernatant at 100× *g* for 4 min at 4 °C. Cultures were processed for RNA isolation one month after harvesting. 

### 4.3. Healthy Donor (HD) Processing

Two tubes of 7.5 mL blood each were drawn from each healthy donor patient and samples were processed as mentioned in the previous section, CTC processing. Of the five healthy individuals, two were male and three were female. Prior to plating of cells for culturing, cells were resuspended in 5 mL 1× PBS. 1 mL of the resuspended solution was extracted for RNA processing. The cells were pelleted at 300× *g* for 3 min at 25 °C and processed into RNA using an RNAqueous-Micro Total RNA isolation kit (Thermo Fisher cat. no. AM1931) according to the manufacturer’s protocol. The remaining cells were plated for culturing. 

### 4.4. RNA Isolation, PCR, qRT-PCR, and Sequencing

Total RNA isolation was performed using RNAqueous-Micro Total RNA isolation Kit (Thermo Fisher cat. no. AM1931) according to manufacturer protocol. RNA was converted to cDNA using LunaScript RT SuperMix Kit (New England Biolabs, Ipswich, MA, USA, cat. no. E3010) according to manufacturer specifications. PCR was performed on cDNA using Hot Start Taq DNA Polymerase (NEB cat. no. M0495) to test for the presence of epithelial (CK8, CK18), mesenchymal (VIM, SERP), and breast markers (MAM). Sequencing of PCR products was conducted via Genewiz (Frederick, MD, USA). Resulting sequencing files were analyzed using SnapGene software (from Insightful Science; available at https://www.snapgene.com/). Quantitative RT-PCR was performed to validate libraries from RNA-sequencing using Luna Universal qPCR MasterMix (New England Biolabs, cat. no. M3003). Primers used for PCR and qRT-PCR can be found in Appendix A. 

### 4.5. RNA-Sequencing

Library preparation and RNA-sequencing from isolated RNA samples was conducted by Genewiz (South Plainfield, NJ, USA) using an Illumina sequencing platform. Only those RNA-samples that yielded a RIN score > 7.0 and yielded sufficient RNA quantity proceeded forward with library preparation. 

### 4.6. Bioinformatics Analysis

Read files were trimmed using Trimmomatic [45] and aligned to the human genome (GRCh38.p13) using the STAR aligner [46]. Aligned reads were quantified using featureCounts [47] and differential expression analysis was performed in R using DESeq2 [48]. Normalized feature counts were used w/ the gene set enrichment analysis, GSEA software and Molecular Signature Database (MSigDB), available at http://www.broad.mit.edu/gsea/. Matlab R2020a was used to generate heatmaps for figures. Colors were scaled by row according to normalized feature counts. Non normalized featured counts were converted to reads per million and input into EPIC (available at http://www.epic.gfellerlab.org/) [28]. CIBERSORT was accessed via http://www.cibersort.stanford.edu/ and non-normalized feature counts were input following developer instructions [29]. A R package for ImSig was accessed at http://www.github.com/ajitjohnson/imsig/, and data was input following developer instructions.

### 4.7. Kaplan-Meier Analysis

Clinical patient data and gene expression data from the METABRIC study were retrieved via http://www.cbioportal.org/. Kaplan–Meier analysis was done using Matlab R2020a. The software X-tile was retrieved from https://medicine.yale.edu/lab/rimm/research/software/ and used to determine a cut point for high and low expression.

### 4.8. Data Availability

RNA-sequencing data and all source data for figures included in the current study have been deposited in the Gene Expression Omnibus (GEO) under accession number GSE148991.

## 5. Conclusions

Ultimately, CTCs represent a unique population of cells that would enable the further study of cancer metastasis. Unfortunately, there remains a lack of efficient and robust methods for the propagation of CTCs. Since CTCs exist in only small numbers within human blood, captured CTCs need to be propagated to be amenable for next-generation sequencing techniques such as RNA-sequencing. Once profiled, CTCs can provide a significant amount of information based solely on their identity as an intermediary stage of metastasis. Comparison of CTCs with primary tumors would enable the identification of metastatic drivers and lead to the development of metastasis-preventing therapies. Thus, having a standardized method for the capture and culture of CTCs is a pressing need.

Despite some of the limitations of the current study, this proof-of-concept provides an important technological advance and fulfills an unmet need. We provide a novel robust and reliable technology for the consistent, un-biased isolation and short-term propagation of CTCs which would provide sufficient cells for elucidating the mechanistic underpinnings of metastasis. Further optimization of the CTC culture will permit us to provide a more granular resolution of CTC heterogeneity at a single-cell level. While previous technologies for CTC culture were encumbered by narrow isolation criteria and/or inadequate culture conditions, the optimized and efficient CTC culture method described here can allow researchers to take advantage of the untapped potential of CTCs.

## Figures and Tables

**Figure 1 cancers-12-02784-f001:**
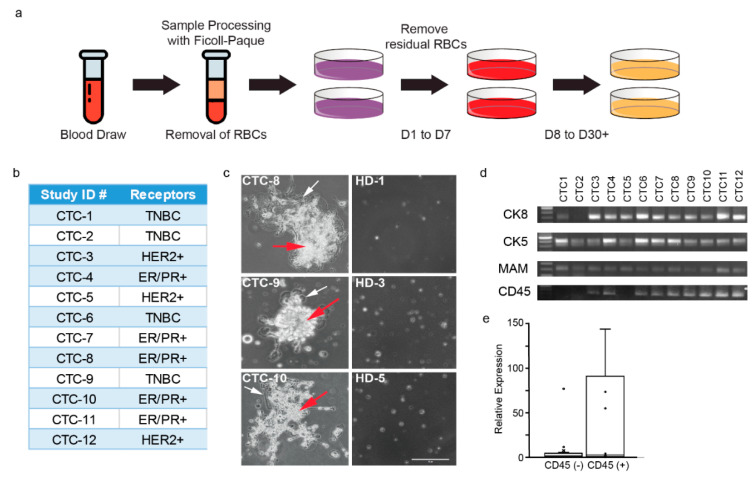
CTC cultures successfully propagated breast epithelial cells. (**a**) Visual workflow of CTC processing. (**b**) Receptor status of cultured CTCs, based on clinical data. (**c**) Representative phase images (20×) taken after 30 days in culture. Within CTC images, red arrows indicate CTCs and white arrows indicate the adherent leukocyte cells on the top of which CTCs attach and expand. (**d**) RT-PCR was used to confirm that RNA isolated from CTCs was of breast and epithelial origin prior to RNA-seq analysis. (**e**) A box and whisker plot depicting CD45 relative expression following qPCR amplification. CD45 (-) samples were observed to have decreased propagation and did not generate enough high-quality RNA for RNA-sequencing analysis (CTCs 1, 2, 5, 6, 7, and 11).

**Figure 2 cancers-12-02784-f002:**
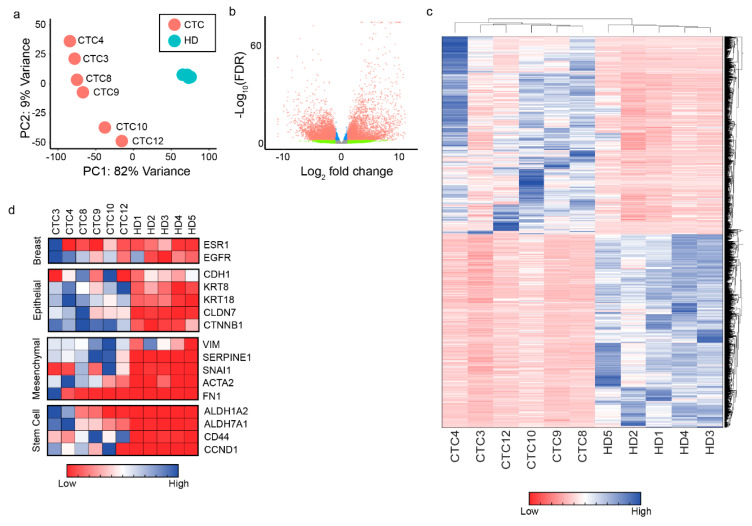
RNA-sequencing data from CTCs demonstrates a distinct “CTC” gene signature. (**a**) Principle-component analysis of RNA-sequencing data reveals that HDs cluster tightly together, while CTCs remain heterogeneous among each other. CTCs cluster closer together than they do to HDs. (**b**) A volcano plot of all significantly differentially expressed genes between CTCs and HDs. (**c**) A clustergram (heatmap) of all significantly differentially expressed genes. (**d**) To further confirm CTCs were of breast and epithelial origin, a select panel of genes were examined in the RNA-sequencing data. Common mesenchymal and cancer stem cell markers were also investigated.

**Figure 3 cancers-12-02784-f003:**
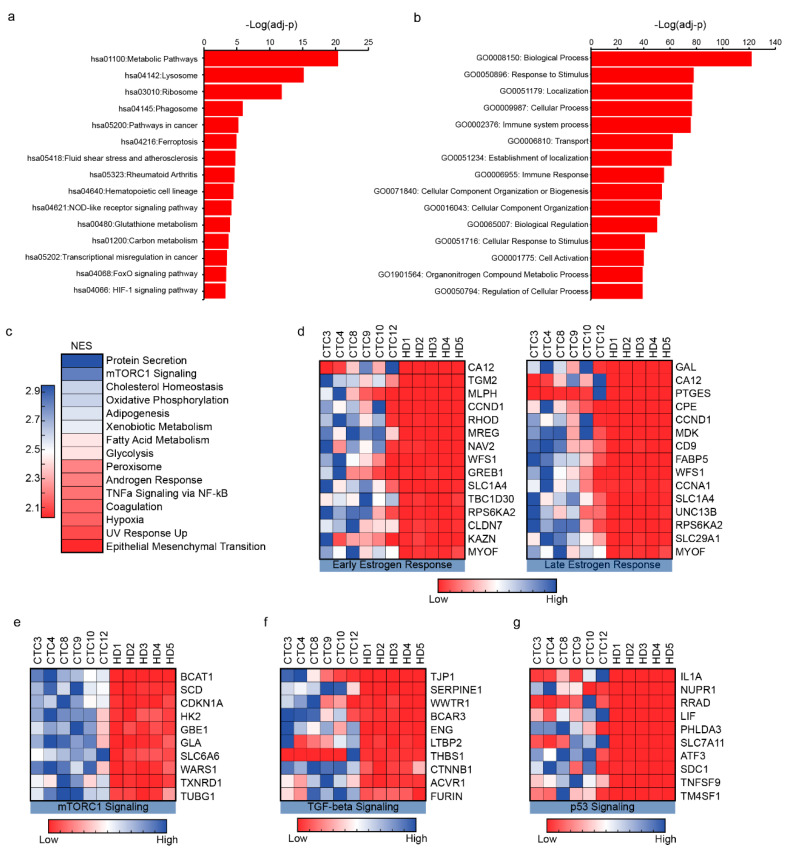
Pathway analysis of RNA-seq data confirms CTC identity, and reveals unique examples of the utility of CTCs. (**a**) KEGG pathway analysis of CTCs. The top 15 pathways (with highest-log (adjusted *p*-values)) are listed here. Notably, cancer associated KEGG pathways, including pathways in cancer (hsa05200) and Table 05202. are revealed. (**b**) The top 15 enriched GO-terms for “Biological Processes” (by *p*-value). Appendix A provides a comprehensive list of all GO-terms enriched. (**c**) GSEA analysis of RNA-sequencing data reveals 31 gene sets were statistically significantly enriched (FDR < 0.25, *p*-value < 0.01). The top 15 gene sets are shown here, colored according to the normalized enrichment score (NES). A higher NES means that genes from that specific gene set are more enriched in CTCs compared to HDs. (**d**) The top 15 genes from two GSEA gene sets, early estrogen response and late estrogen response, are shown. The enrichment of these gene sets confirms that the system likely isolated cells of breast origin. (**e**–**g**) The top 10 upregulated genes from selected GSEA gene sets are shown.

**Figure 4 cancers-12-02784-f004:**
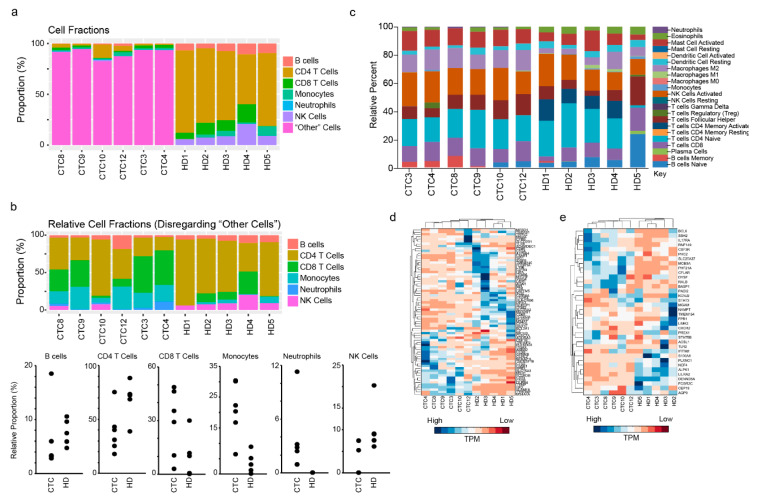
Cultures propagated CTCs and favored the survival of neutrophils. (**a**) EPIC output of cell types based on RNA-sequencing data. The “other” category refers to any non-immune cells (of which CTCs would be a part of). (**b**) EPIC deconvolution of cell types, without considering the “other” category. This allows for the closer inspection of immune cell types present according to enriched RNA transcripts. (**c**) CIBERSORT deconvolution results of immune cell types. CIBERSORT utilizes a different method for the identification of immune cells than EPIC. (**d**) A clustergram of the *ImSig* macrophage signature. 31/71 genes were enriched in CTCs compared to HDs. (**e**) A clustergram of the *ImSig* neutrophil signature. 17/35 genes were enriched in CTCs compared to HDs.

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
