# Peer review of "Efficient Propagation of Circulating Tumor Cells: A First Step for Probing Tumor Metastasis"

_cancers, 2020, doi:10.3390/cancers12102784_

Round 1

Reviewer 1 Report

Comment for authors

The article ‘Efficient propagation of circulating tumor cells: a first step for probing tumor metastasis’ reported a method for culturing the circulating tumor cells from breast cancer patients and further characterized these CTC cells with RNA-seq. The author investigated an important role of CTC with proper methodologies and research design, but functional data is missing. This reviewer has the following comments.

  1. Methods section reported that the study includes both females and males. Does this study really use any samples/blood from male breast cancer patients?
  2. Why only 50% (/12) patients’ cells only grow this new culture method or CTC isolation technique? Is this samples or patients’ characteristics are deciding factor for whether cells will grow or not, nothing to with methods?
  3. What are the other factors or differences, other CD45, in the sample which are enable those particular samples?
  4. Why 30 days of culture time chosen? These cells do not grow reasonably fast, which they are cancer/metastasis clones?
  5. Figure-1d: do authors tested HD samples in RT-PCE for the same markers? How authors are sure the RNA after 30 days culture are only from the CTC, not other cells or may some other population differentiate to CTC-like phenotype. Need run HD samples/RNA in PCR.
  6. Figure-4: authors mentioned that no other cells are survived in their culture conditions, but still they see signature gens for most immune cells. How does it possible or where these signatures of other populations coming? Does some cell still survive in culture?
  7. Why HD RNA data did not show any neutrophil signature, even RNA is isolated from buffy coat? It should have neutrophils in it.
  8. Authors need to verify some of the key genes/markers for CTCs signature from RNA-seq by other methods like flow cytometry or qPCR or some functional test are needed.
  9. There is a lack of defining the most of abbreviation at its first appearance.

Reviewer 2 Report

This finding is revolutionary, however, the methods must be described in more detail. How were the red blood cells removed? Lysis? Please describe in detail. All the media additives are not quantified. This must be done in order to be able to reproduce the data.

Round 2

Reviewer 2 Report

Corrections well done